# Lactate Activates AMPK Remodeling of the Cellular Metabolic Profile and Promotes the Proliferation and Differentiation of C2C12 Myoblasts

**DOI:** 10.3390/ijms232213996

**Published:** 2022-11-13

**Authors:** Yu Zhou, Xi Liu, Caihua Huang, Donghai Lin

**Affiliations:** 1Key Laboratory for Chemical Biology of Fujian Province, MOE Key Laboratory of Spectrochemical Analysis and Instrumentation, College of Chemistry and Chemical Engineering, Xiamen University, Xiamen 361005, China; 2Research and Communication Center of Exercise and Health, Xiamen University of Technology, Xiamen 361024, China

**Keywords:** lactate, metabolic regulator, NMR-based metabonomics, AMPK, C2C12 myoblasts

## Abstract

Lactate is a general compound fuel serving as the fulcrum of metabolism, which is produced from glycolysis and shuttles between different cells, tissues and organs. Lactate is usually accumulated abundantly in muscles during exercise. It remains unclear whether lactate plays an important role in the metabolism of muscle cells. In this research, we assessed the effects of lactate on myoblasts and clarified the underlying metabolic mechanisms through NMR-based metabonomic profiling. Lactate treatment promoted the proliferation and differentiation of myoblasts, as indicated by significantly enhanced expression levels of the proteins related to cellular proliferation and differentiation, including p-AKT, p-ERK, MyoD and myogenin. Moreover, lactate treatment profoundly regulated metabolisms in myoblasts by promoting the intake and intracellular utilization of lactate, activating the TCA cycle, and thereby increasing energy production. For the first time, we found that lactate treatment evidently promotes AMPK signaling as reflected by the elevated expression levels of p-AMPK and p-ACC. Our results showed that lactate as a metabolic regulator activates AMPK, remodeling the cellular metabolic profile, and thereby promoting the proliferation and differentiation of myoblasts. This study elucidates molecular mechanisms underlying the effects of lactate on skeletal muscle in vitro and may be of benefit to the exploration of lactate acting as a metabolic regulator.

## 1. Introduction

As the largest metabolic organ in the human body, skeletal muscle accounts for about half of the body mass [1,2]. Maintenance of the homeostasis of skeletal muscle is crucial to the health of the body. Satellite cells (SCs) of skeletal muscle fibers have the potential of being differentiated into muscle fibers [3,4], laying the basis for injury repair and the regeneration of adult skeletal muscle [5,6,7]. During homeostasis, SCs are in a “resting state” with a relatively inactive metabolism. When stimulated by stress, injury, or others, SCs are transformed into an “activated state”. The activated SCs are continually proliferated and differentiated, generating new muscle fibers. Previous studies showed that myogenic regulatory factors (MRFs) [8,9] including MyoD, myogenin, Myf5 and MRF4, can regulate expressions of skeletal muscle-specific genes, thus playing key roles in myogenic differentiation. The expressions of MyoD and Myf5 are promoted in the activated SCs, then cell differentiation is promoted, accompanied by enhanced expressions of myogenin and MRF4. The expression level of myosin heavy chain (MyHC) serving as the main component of structural muscle-specific protein, reflects the formation of myotubes [10,11,12].

Compared with the resting SCs, the activated SCs increase energy demand, promote protein synthesis, and activate relevant signal pathways, such as PI3K/Akt and MAPK pathways. During muscle differentiation, expressions of p38 and ERK1/2 (extracellular signal-regulated kinase 1/2) are significantly regulated, thereby promoting the proliferation and differentiation of SCs [13,14,15,16]. The PI3K/Akt pathway is activated during muscle hypertrophy, but inhibited during muscle atrophy, as shown in the models of skeletal muscle hypertrophy and atrophy in vivo [17]. As reported previously, the “resting state” and “activated state” of satellite cells showed significantly different metabolic profiles [18]. Moreover, activated SCs exhibit promoted the expressions of genes associated with glycolysis and the TCA cycle, as indicated by a single-cell RNA-sequencing study on satellite cells isolated from a skeletal muscle mouse injury model [19]. Furthermore, a previous study showed that AMPK acting as a core molecule in metabolic regulation can regulate the activation of satellite cells by controlling metabolic homeostasis [20]. It has been demonstrated that the non-canonical Sonic hedgehog (Shh) pathway promotes the proliferation of SCs by activating AMPK, which is attenuated by AMPK-specific knockdown [21]. Additionally, glutamine intake in SCs via the glutamine transporter SLC1A5 can promote mTOR signaling and enhance the proliferation and differentiation capabilities of SCs, thereby boosting muscle regeneration [22]. These results suggest that metabolic regulation is closely related to the activation of SCs.

As a major circulating carbohydrate fuel [23], lactate is mostly produced from anaerobic muscles under some conditions such as exercise and muscle damage [24]. In the past years, lactate had been considered a metabolic waste product, but more recently, lactate has been generally recognized as an important energy substrate and signaling molecule [23,25]. As is known, pyruvate is converted to lactate under the enzymatic catalysis of lactate dehydrogenase isoform A (LDHA), which is contrarily converted to pyruvate under the catalysis of lactate dehydrogenase isoform B (LDHB). In the absence of lactate dehydrogenase, glycolysis-produced pyruvate and NADH are always metabolized via the mitochondrial TCA cycle [26,27]. Lactate dehydrogenase uncouples glycolysis and the TCA cycle metabolism, the efficiency of which depends on the expressions of monocarboxylate transporters MCT1/4 and the molecular chaperone CD147 [25,26,28]. Since most mammalian cells can express LDH (lactate dehydrogenase) and MCTs [29,30,31], both glycolysis and TCA cycle metabolism work independently. A previous study has shown that lactate produced from the glycolysis of endothelial cells can regulate macrophage metabolism, induce macrophage polarization, and promote muscle angiogenesis in ischemic-injured muscle [32]. In addition, lactate can also regulate myogenesis [33,34] and increase the myotube diameter [35], and stimulate skeletal muscle hypertrophy [36]. Nevertheless, it remains unclear whether lactate can regulate the metabolism of muscle cells.

Here, we assessed the effects of lactate treatment on the proliferation and differentiation of C2C12 myoblasts and exploited the underlying metabolic mechanisms by conducting NMR-based cellular metabolomic analysis. Our results showed that lactate can distinctly remodel cellular metabolic profiles as a metabolic regulator, and significantly promote the proliferation and differentiation of myoblasts.

## 2. Results

### 2.1. Lactate Treatment Promoted the Proliferation of Myoblasts

The repair and regeneration of myofibers depend on the proliferation and myogenic differentiation of muscle satellite cells during muscle remodeling [3,6]. We first evaluated the effect of lactate treatment on the proliferation of myoblasts. Considering the physiological range of lactate concentration in blood is around 5 mM after a certain time of mid-intensity aerobic exercise in humans, we chose 5 mM as the concentration of lactate for treating C2C12 cells [37]. The C2C12 cells were cultured for 24 h in either DMEM medium (Con) or DMEM medium supplemented with 5 mM lactate (Lac). Significantly, lactate treatment enhanced the proliferation capability of the cells (Figure 1A,B) and increased cell viability by 25% (Figure 1C). As shown previously, both the AKT pathway and ERK pathway can induce muscle protein synthesis serving as common anabolic signaling [16,17], and can stimulate the proliferation and differentiation of myoblasts [35]. Consistently, we found that lactate treatment significantly enhanced the expression levels of p-AKT and p-ERK in the cells (Figure 1D–F), thereby promoting protein synthesis in muscle satellite cells and the proliferation of myoblasts.

### 2.2. Lactate Treatment Promoted the Myogenic Differentiation of Myoblasts

We then evaluated the effect of lactate treatment on the myogenic differentiation of myoblasts. The four-day treatment with lactate obviously promoted the fusion of myoblasts (Figure 2A), as shown by a 19% increase in the fusion index of the lactate-treated cells (Figure 2B,C). This result indicated that lactate treatment significantly promoted the myogenic differentiation of myoblasts.

### 2.3. Lactate Treatment Promoted Expressions of Myogenic Regulatory Factors and Myosin Heavy Chain

As reported previously, p38-MAPK signaling induces the expression of p21, which allows the cell cycle and differential process for myoblasts, and also assists MyoD in activating expressions of target genes [15]. MyoD, Myogenine, Myf-5 and MRF4 (also called Myf6) are the main members of the myogenic regulatory factors (MRFs) [38,39], in which Myogenin and MyoD can strictly regulate the formation of myotubes [8]. Significantly, lactate treatment upregulated expressions of MyoD and MyoG in myoblasts, but did not observably regulate the phosphorylation of p38 (Figure 3). Furthermore, the expression level of the contractile protein myosin heavy chain (MYHC) was profoundly elevated in lactate-treated myotubes. Usually, MYHC is highly expressed in muscle, and its expression level represents the maturation of a myotube [40]. These results supported the lactate-induced promotion of the myogenic differentiation of myoblasts.

### 2.4. Lactate Treatment Remodeled the Metabolic Profile of Myoblasts

Typical one-dimensional (1D) ^1^H spectra were recorded on aqueous metabolites derived from the Lac and Con groups of cells (Figure 4). In total, 37 metabolites were identified (Appendix A) and the resonance assignments of the metabolites were confirmed by two-dimensional (2D) ^1^H-^13^C HSQC spectra (Appendix A).

The NMR signal of lactate was excluded from the multivariate data analysis to eliminate the potential influence of the supplemented lactate on the metabolic profile of lactate-treated myoblasts. We established the principal component analysis (PCA) model to unbiasedly observe the systemic metabolic pattern of myoblasts. The PCA scores plot exhibits the metabolic separation between the Lac and Con groups of cells (Figure 5A), implying that lactate treatment obviously remodeled the metabolic profile of myoblasts. 

To maximize the distinction of the metabolic profiles between the Lac and Con groups of cells, we established the partial least squares discriminant analysis (PLS-DA) model. The PLS-DA scores plot displays the metabolic distinction between the two groups (Figure 5B). The response permutation test (200 cycles) was executed to assess the reliability of the established PLS-DA model. The resultant cross-validation plot is indicative of the high reliability of the PLS-DA model with favorable abilities of explanation and prediction (Appendix A).

### 2.5. Lactate Treatment Changed the Levels of Metabolites in Myoblasts

Relative levels of the assigned 37 metabolites in myoblasts were measured based on NMR integrals of the metabolites and TSP (Appendix A). The established PLS-DA model was used to screen significant metabolites with variables important in projection (VIP) > 1, which primarily contributed to the metabolic distinction between the Lac and Con groups of cells. In total, eight significant metabolites were identified, including myo-inositol, alanine, methanol, sn-glycero-3-phosphocholine, glutamine, glycine, phosphocreatine and glutamate (Figure 5C). Furthermore, a univariate analysis was conducted to compare the relative levels of the metabolites between the two groups. Nine differential metabolites with *p* < 0.05 were identified, including alanine, glutamate, succinate, NAD^+^, myo-inositol, glycine, phosphocreatine, glutamine and sn-glycero-3-phosphocholine (Figure 6). Except for succinate, these differential metabolites were significantly increased in the Lac group relative to the Con group. 

By a combination of the significant metabolites and differential metabolites, we identified seven characteristic metabolites, including myo-inositol, alanine, sn-glycero-3-phosphocholine, glycine, glutamine, glutamate and phosphocreatine.

### 2.6. Lactate Treatment Altered the Metabolic Pathways in Myoblasts

The relative levels of metabolites were used to screen significantly altered metabolic pathways (termed as important metabolic pathways thereafter) with two criteria of pathway impact value (PIV) > 0.1 and *p* < 0.05. Compared with the Con group, the Lac group showed five important metabolic pathways (Figure 7 and Appendix A): (1) Alanine, aspartate and glutamate metabolism; (2) D-Glutamine and D-Glutamate metabolism; (3) glyoxylate and dicarboxylate metabolism; (4) arginine biosynthesis; and (5) glutathione metabolism.

### 2.7. Lactate Treatment Upregulated the Energy Metabolism in Myoblasts

As it is known, lactate can not only act as a substrate molecule providing energy for organs and tissues via an enzymatic reaction but also serve as a signal molecule mediating physiological and pathological changes in the body [23,25]. Both the identified important metabolic pathways and significant metabolites suggested that lactate treatment might enhance the proliferation and differentiation capabilities of myoblasts by regulating amino acid metabolism and promoting energy metabolism. Due to the core role of AMPK signaling for both regulating energy metabolism of the body and activating satellite cells, we further analyzed the effects of lactate treatment on the expression levels of crucial proteins involved in the AMPK signaling pathway, including p-AMPK and p-ACC (acetyl-CoA carboxylase). Obviously, lactate treatment enhanced the expression level of p-AMPK, indicating promoted AMPK signaling (Figure 8A,B). The inhibition of AMPK signaling using dorsomorphin (an inhibitor of AMPK) significantly blocked the promotive effect of lactate treatment on C2C12 cells (Appendix A). To our knowledge, this represents the first finding that lactate treatment can activate the AMPK in muscle satellite cells.

Moreover, lactate treatment also raised the expression level of p-ACC (Figure 8A,C), implying an increased contribution of acetyl-CoA to TCA cycle metabolism. Furthermore, lactate treatment significantly promoted the expressions of the monocarboxylate transporter (MCT1) and its chaperone protein CD147 (Figure 8A,D,E), and also increased the enzymatic activity of LDHB catalyzing the enzymatic reaction of converting lactate to pyruvate (Figure 8H). Expectedly, these lactate-induced effects favored the TCA cycle energy metabolism in myoblasts. The allosteric regulation of lactate might be responsible for these effects. Nevertheless, future works should be conducted to clarify the molecular mechanisms of lactate treatment promoting the expressions of p-AMPK, p-ACC, MCT1 and CD147 as well as the enzymatic catalysis of LDHB. 

In summary, lactate treatment profoundly upregulated the metabolism in myoblasts through activating AMPK, including promoting the intake and intracellular utilization of lactate, enhancing the expression level of p-ACC, activating the TCA cycle and creatine–phosphocreatine system, thereby increasing energy production and promoting the proliferation and differentiation of myoblasts. To overview these results, we constructed a schematic representation to summarize the metabolic mechanisms accounting for the effects of lactate treatment on the proliferation and differentiation of myoblasts based on KEGG metabolic pathways (Figure 9).

## 3. Discussion

Lactate plays crucial roles in many metabolic processes acting as an important energy substrate and key node of metabolic regulation connecting glycolysis and oxidative phosphorylation in cells, tissues and organs [26]. A great quantity of lactate is usually produced in skeletal muscles, inducing the adaption of skeletal muscle during strenuous exercise [41,42]. The underlying molecular mechanisms remain elusive. As it is known, the regeneration and maintenance of skeletal muscle could be directly affected by muscle satellite cells [39,43]. Extensively, myoblasts are used to establish muscle satellite cell models since they can be induced by myogenic differentiation in vitro. In this research, we exploited the effects of lactate on myoblasts to mimic the response of muscle satellite cells to lactate during exercise, aiming to clarify potential metabolic mechanisms underlying the adaption of skeletal muscle. 

Skeletal muscles serve as important locomotive organs and dominant energy metabolic centers. The energy of the metabolic homeostasis of skeletal muscles could affect the physiological balance of the whole body [1]. Recent studies have shown that lactate activates the ERK1/2 and AKT pathways to varying degrees in different muscle types [44], and promotes hypertrophy and the regeneration of skeletal muscle [36]. Furthermore, muscle satellite cells could directly affect the regeneration and maintenance of skeletal muscle [39,43]. In this study, we found that the cell viability of C2C12 myoblasts was significantly increased by lactate treatment. Cell proliferation always requires sufficient protein synthesis, which can be activated by the AKT pathway [17]. Consistent with previously published studies, the present study observed the upregulated expression of p-ERK, suggesting that lactate supplementation favors the myogenic differentiation of myoblasts and increases the myotube diameter [35]. It was noted that the concentration of lactate (20 mM) used in the previous study [44] differed from that used in our study. Considering the overloading effect of lactate in skeletal muscle [34], here we chose 5 mM as the concentration of lactate for treating C2C12 cells. Obviously, the upregulated expression levels of MyoD, Myogenin and MYHC reflected the substantially enhanced differential capability of the cells treated by lactate at such a lower concentration.

Additionally, changes to life activities would affect metabolic homeostasis, and in turn, metabolic disturbance might also influence life activities [23,45,46]. We observed that lactate treatment induced profound metabolic alterations in myoblasts. The identified characteristics showed that metabolites were substantially responsible for the lactate-induced metabolic changes, including phosphocreatine, glutamate, glutamine, glycine, alanine, myo-inositol and glycerophosphocholine. In summary, our results suggest that lactate treatment significantly promotes the proliferation and myogenic differentiation of cells probably through the remodeling of the cellular metabolic profile. The metabolic remodeling is primarily contributed to by the promoted intake and intracellular utilization of lactate and activated the TCA cycle in lactate-treated myoblasts. 

### 3.1. Lactate Treatment Promotes the Intake and Intracellular Utilization of Lactate

As the storage form of glucose in skeletal muscle, glycogen can release glucose under the catalysis of glycogen phosphorylase in periods of fluctuating energy supply. The released glucose can be transformed into pyruvate, then producing lactate and NADH through anaerobic glycolysis [47]. The mutual transformation of lactate and pyruvate can occur under the catalysis of LDH in certain circumstances. In detail, LDHA promotes the conversion from pyruvate to lactate in high glycolytic cells, while LDHB favors the conversion of pyruvate into lactate in high oxidative cells. In this study, lactate treatment distinctly enhanced the capacity of lactate intake and the enzymatic activity of LDHB, thus promoting the intracellular utilization of lactate in myoblasts (Figure 8). As reported previously, lactate can shuttle among cells via monocarboxylic acid transporters (MCTs), participating in the energy supplementation of important tissues and organs such as skeletal muscle, liver and brain [27,48,49,50]. Expectedly, the promoted intracellular utilization of lactate would provide more substrates for the TCA cycle in favor of energy production.

### 3.2. Lactate Treatment Activates TCA Cycle by Regulating Amino Acid Metabolism

Our previous studies showed that both α-ketoglutarate supplementation and the AG dipeptide supplement can improve the energy deficiency of myoblasts cultured in low-glucose media by activating the TCA cycle. These supplements significantly increased several amino acids which were distinctly decreased by energy deficiency, including alanine, glycine, glutamate and glutamine [46,51]. Alanine can be converted into pyruvate through a series of enzymatic reactions, and can then participate in the TCA cycle through the decarboxylation of pyruvate to produce acetyl-CoA and NADH [52]. Similarly, glutamate and glutamine can be converted to α-ketoglutarate, finally entering the TCA cycle. Additionally, it was previously reported that the deficiency of glutamine could reduce the proliferation and differentiation capabilities of myoblasts cultured in glutamine-free media [53]. Here, we detected elevated levels of alanine, glutamate and glutamine in lactate-treated myoblasts (Figure 6), implying that lactate treatment could activate the TCA cycle. In addition, we measured the enzyme activities of several enzymes including isocitrate dehydrogenase (IDH), malate dehydrogenase (MDH) and succinate dehydrogenase (SDH), and found that the enzymatic activities of IDH and MDH were observably increased (Appendix A). These results further indicate the lactate-induced effects of promoting cellular energy metabolism, thereby promoting the proliferation and differentiation of myoblasts. 

### 3.3. Lactate Treatment Promotes Energy Production in Myoblasts

The creatine–phosphocreatine–creatine kinase (Cr/PCr/CK) system is one of the main metabolic systems responsible for producing ATP in skeletal muscles [54]. As the energy currency in the body, ATP is extensively used in various cellular and indispensable processes for the maintenance of cell homeostasis. In the Cr/PCr/CK system, PCr can be applied to generate ATP via the catalysis of CK for cellular life activities, which is fundamental in the rapid re-synthesis of ATP. In this study, lactate treatment significantly increased PCr, indicating that the exogenous supplementation of lactate increases the ATP production in myoblasts for meeting energy demands in the proliferation and differentiation of the cells (Figure 8). Further work is required to directly characterize the promoted energy production in lactate-treated myoblasts.

### 3.4. Lactate Treatment Activates AMPK Promoting the Proliferation and Differentiation of Myoblasts

Lactate upregulates the expressions of genes related to oxidative metabolism and promotes mitochondrial biogenesis in L6 cells [55] and skeletal muscle [56]. Interestingly, Cerda-Kohler and colleagues showed that lactate stimulation promoted the phosphorylation of AMPK in the soleus, but did not induce observable changes in the extensor digitorum longus [44]. AMPK usually acts as a prime regulator of metabolic homeostasis, and the activation of AMPK would promote the energy metabolism of the TCA cycle [20,45,57]. This suggests the role of lactate in the metabolic regulation of skeletal muscle. In this study, the lactate-treated myoblasts exhibited an obviously activated AMPK signal pathway, and the enhanced expression of p-ACC further confirmed that lactate treatment promotes AMPK signaling (Figure 8A–C). Thus, lactate might act as a metabolic regulator for promoting the energy metabolism of the TCA cycle by activating the AMPK in myoblasts. These results might lay the molecular basis for the lactate-induced effects of enhancing the proliferation and differentiation capabilities of myoblasts.

## 4. Materials and Methods

### 4.1. Cell Culture 

The murine myoblast cell line C2C12 was purchased from National Biomedical Experimental Cell Resource Bank (Beijing, China). Myoblasts were grown at 37 °C in DMEM (high glucose, HyClone) culture medium containing 10% fetal bovine serum (Biological Industries), 100 units/mL penicillin, and 100 μg/mL streptomycin in a humidified atmosphere (5% CO_2_). When the cells reached 85–90% confluence, the differentiation of myoblasts was induced to form myotubes by changing the culture medium to DMEM medium containing 2% horse serum, 100 units/mL penicillin and 100 μg/mL streptomycin. Fresh differentiation medium was changed daily until the cells were harvested for analysis. Sodium lactate (#71718, Sigma–Aldrich, Shanghai, China) was added in either the culture medium or the differentiation medium at a final concentration of 5 mM. The observation of morphologies characteristics was carried out with an inverted microscope at 20× magnification (Motic, AE31E, Xiamen, China).

The cells cultured in the DMEM media with and without lactate supplementation were defined as the Lac and Con groups of myoblasts, respectively. Similarly, the cells differentiated in the differentiation media with and without lactate supplementation were defined as the Lac and Con groups of myotubes, respectively.

### 4.2. Cell Viability Assay

Myoblasts were seeded in 96-well plates and cultured for 24 h, then the culture media were replaced by fresh media supplemented with 5 mM lactate or 10 μM dorsomorphin or 5 mM lactate + 10 μM dorsomorphin for culturing for another 24 h. Thereafter, 10% MTS (Cell Titer 96 Aqueous solution, Promega, Madison, WI, USA) was added to each well, and the absorbance at 490 nm was detected with a multimode microplate reader (BioTek, Winooski, VT, USA). 

### 4.3. Immunofluorescence Staining and Imaging

Myotubes were fixed with prechilled 4% paraformaldehyde (30 min), punched in 0.4% Triton (10 min) and then blocked in 5% BSA (60 min). Thereafter, the myotubes were incubated with anti-MYHC (sc-376157, Santa Cruz Biotechnology, Dallas, TX, USA) antibody overnight, washed three times with PBS, and incubated (60 min) with goat anti-mouse IgG conjugated with FITC (Biowarld, Beijing, China). The nuclei were stained with DAPI (Sigma) after washing three times with PBS. Images of the myotubes were taken using a fluorescence microscope (Motic, Xiamen, China).

### 4.4. Enzyme Activities Assay

C2C12 cells were collected using extracting solution (5 × 10^6^ cells: 1 mL) and disrupted under an ice bath by ultrasonic (200 W, continuous 3 s in every 10 s, repeat 30 times). The supernatant was separated after centrifuging (8000× *g*, 10 min, 4 °C). The enzymatic activity of LDH, IDH, MDH and SDH was measured using commercial assay kits purchased from Beijing Solarbio Technology Co., Ltd. (Beijing, China) under the guidance of the manufacturer’s protocol. 

### 4.5. Western Blotting

Proteins contained in myoblasts or myotubes were extracted using RIPA lysis buffer (Solarbio Technology Co., Ltd. Beijing, China), then the lysates were sonicated (1 min) and centrifuged (12,000× *g*, 15 min, 4 °C). SDS loading buffer was added to the supernatant after the total protein concentration was determined using a BCA protein assay kit (Lab Lead, Xiamen, China). Then, proteins (10–30 μg) were separated on SDS-PAGE (10–15%) and transferred onto PVDF membranes (GE Healthcare, Shanghai, China) for immunoblotting analysis. After being blocked (60 min) with 5% BSA (Beyotime Biotechnology, Shanghai, China) at room temperature, the membranes were incubated at 4 °C with corresponding primary antibodies overnight, including AKT (Proteintech, Wanhan, China), p-AKT (ABGENT), p38 (Proteintech), p-p38 (ABGENT, San Diego, USA), ERK (Cell Signaling, Danvers, MA, USA ), p-ERK (Cell Signaling, Danvers, MA, USA), and GAPDH (Proteintech, Wuhan, China). MyoD (Santa Cruz Biotechnology, Dallas, TX, USA), Myogenin (Santa Cruz Biotechnology, Dallas, TX, USA), ACC (Cell Signaling, Danvers, MA, USA), p-ACC (Cell Signaling, Danvers, MA, USA), AMPK (Cell Signaling, Danvers, MA, USA), p-AMPK (Cell Signaling, Danvers, MA, USA), CD147 (Proteintech, Wuhan, China), MCT1 (Proteintech, Wuhan, China) and LDHB (Proteintech, Wuhan, China), followed by horseradish peroxidase-conjugated secondary antibody for 1 h at room temperature. The visualization of protein was conducted using an enhanced chemiluminescence reagent (ECL, Beyotime Biotechnology) and imaged with ChemiScope Capture and analyzed by ChemiScope Analysis software (ChemiScope 6000, CLiNX, Shanghai, China). 

### 4.6. Extraction of Aqueous Metabolites and NMR Sample Preparation

Aqueous metabolites were extracted from myoblasts according to the liquid–liquid extraction method described previously [58]. Briefly, cells (about 1 × 10^7^/petri dish) were harvested after washing three times using prechilled PBS (pH 7.4). A cocktail solvent mixture of methanol, chloroform and water (4:4:2.85) was added to the cells. Methanol was removed from the upper extracts by nitrogen blowing and dried by Freeze dryer (LGJ-10E, Foring Technology Development, Beijing Co., Ltd. Beijing, China) under −65 °C and Nitrogen blowing at 20 Pa for 24 h and suspended in 550 μL of 50 mM phosphate buffer (100% D_2_O, 0.05 mM TSP, pH 7.4). Here, TSP was used as an internal standard for both calibrating the chemical shifts of NMR spectra and quantitatively measuring the levels of the metabolites. The redissolved samples were transferred into NMR tubes and centrifuged prior to the experiments.

### 4.7. NMR Experiments

All detections were conducted on a Bruker Avance III HD 850 MHz spectrometer equipped with a TCI cryoprobe (Bruker BioSpin, Germany) at 25 °C. One-dimensional ^1^H spectra were acquired using the standard pulse sequence NOESYGPPR1D [RD-G1-90°-t1-90°-τm-G2-90°-ACQ]. The water resonance was suppressed during the relaxation delay and mixing time, and pulsed gradients G1 and G2 were used to improve the quality of water suppression. The following experimental parameters were used: spectral width: 20 ppm, acquisition time: 2.66 s, relaxation delay: 4 s and transients: 64.

### 4.8. NMR Data Processing

Fourier transformation was executed after applying an exponential function with a 0.3-Hz line-broadening factor to free induction decay (FID) signals. Subsequently, phase correction, baseline correction and resonance alignment were conducted on all 1D ^1^H spectra with the MestReNova 9.0 software (Mestrelab Research S.L., Santiagode Compostela, Spain). Chemical shifts of the NMR spectra were referenced to the methyl resonance of TSP (δ 0.00). Spectral regions of δ 4.85–4.75 (water resonance) were removed from the spectra, then the regions of δ 9.5–0.75 were binned by 0.001 ppm in MATLAB R2014b (MathWorks, Natick, USA) to obtain a data matrix for the following multivariate analysis. For each spectrum of a given NMR sample, peak integrals were normalized based on both the peak integral of TSP and the number of cells. 

### 4.9. NMR Resonance Assignments

The resonances of metabolites were assigned by Chenomx NMR Suite 8.3 software (Chenomx Inc, Edmonton, Canada) combined with the Human Metabolome Database and references [59,60]. Two-dimensional ^1^H-^13^C HSQC spectra were required for the confirmation of resonance assignments.

### 4.10. Multivariate Statistics Analysis and Identification of Significant Metabolites

Multivariate analysis was performed on the NMR dataset through the SIMCA-P 14.1 software (Umetrics, Umea, Sweden). The non-supervised PCA was conducted to illustrate metabolically grouping treads of myoblasts and show clustering for the NMR dataset. Then, the supervised PLS-DA was carried out to maximally discriminate the metabolic profiles between the Lac and Con groups of cells. The response permutation test was executed to assess the liability of the PLS-DA model by calculating the interpretive and predictive abilities. Significant metabolites were confirmed with a criterion of variable importance in projection (VIP) > 1 after ensuring the favorable interpretation and prediction of the established PLS-DA model.

### 4.11. Univariate Statistics Analysis and Identification of Differential Metabolites

Both relative integrals of singlet or nonoverlapped peaks in spectra and proton numbers in metabolites were used to calculate relative levels of the metabolites, which are presented as mean ± standard deviation (SD). Student’s *t*-test was executed to identify differential metabolites with a criterion of *p* < 0.05 by using the SPSS 19 software.

The combination of differential metabolites and significant metabolites gave characteristic metabolites with a (VIP) > 1 and *p* < 0.05.

### 4.12. Metabolic Pathway Analysis and Identification of Crucial Metabolic Pathways

Based on the relative levels of the identified metabolites, significantly altered metabolic pathways were determined with two criteria of *p* < 0.05 (namely −lg*p* > 1.3) and PIV > 0.2 using the MetaboAnalyst 5.0 software. Both *p* and PIV values were calculated from the metabolite set enrichment analysis and pathway topological analysis, respectively.

### 4.13. Statistical Analysis

Student’s *t*-test was conducted to quantitatively compare the Lac and Con groups of cells with the GraphPad Prism 8.0.2 software (La Jolla, San Diego, CA, USA). Experimental data were presented as means ± SD. Statistical significances were shown as follows: *p* > 0.05 (NS), *p* < 0.05 (*), *p* < 0.01 (**), *p* < 0.001 (***), and *p* < 0.0001 (****).

## Figures and Tables

**Figure 1 ijms-23-13996-f001:**
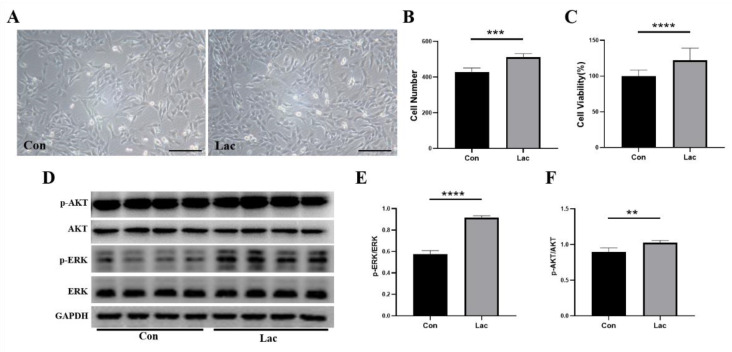
Lactate treatment promoted the proliferation of myoblasts. Cells were cultured for 24 h in either DMEM medium (Con) or DMEM medium supplemented with 5 mM lactate (Lac). (**A**) Representative morphological images of the cells. Scale bar: 200 μm; (**B**) Cell number statistics in panel (**A**) (*n* = 5); (**C**) Viabilities of lactate-treated cells relative to controls measured by the MTS assay (*n* = 10); (**D**–**F**) Expressions of p-AKT (Ser473), AKT, p-ERK (Thr202/Tyr204) and ERK in the Lac and Con groups of cells (*n* = 4). Results are presented as Mean ± SD. ** *p* < 0.01, *** *p* < 0.001, **** *p* < 0.0001 for the statistical analysis of Lac vs. Con.

**Figure 2 ijms-23-13996-f002:**
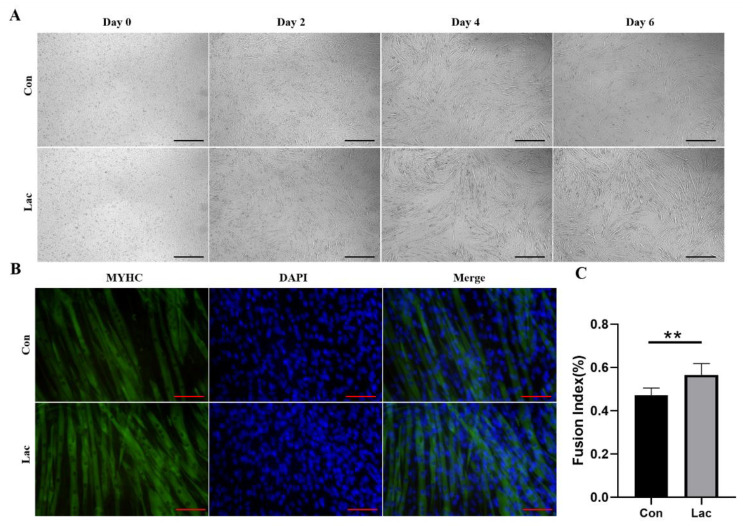
Lactate treatment promoted the myogenic differentiation of myoblasts. Cells were differentiated in either a differentiation medium (Con) or a differentiation medium supplemented with 5 mM lactate (Lac). (**A**) Representative morphological images of cells formed after inducing differentiation for the indicated days. Scale bar, 500 μm; (**B**) Lactate-treated cells and controls formed 7 days after the induction of differentiation were stained with MYHC antibody (green) and DAPI (blue). Scale bar, 20 μm; (**C**) Fusion index defined as the percentage (%) of nuclei located within MYHC-positive myotubes out of the total number of nuclei (*n* = 4). Results are presented as Mean ± SD. ** *p* < 0.01, for the statistical analysis of Lac vs. Con.

**Figure 3 ijms-23-13996-f003:**
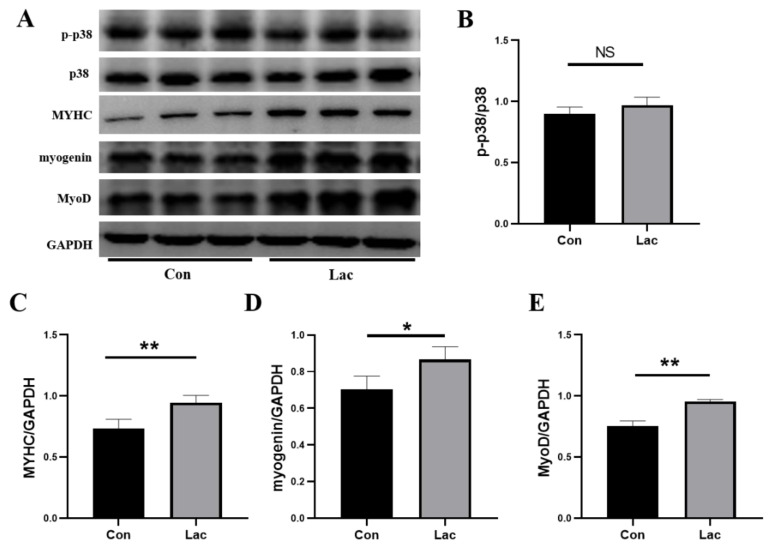
Lactate treatment promoted expressions of myogenic differentiation-related proteins in C2C12 myotubes. Myoblasts were differentiated either in a differentiation medium (Con) or a differentiation medium supplemented with 5 mM lactate (Lac). (**A**) Expressions of p-p38 (Tyr182), MyoD, myogenin, and MYHC in Lac and Con groups were analyzed by Western blot; (**B**–**E**) Gray value analyses corresponding to Panel (**A**) (*n* = 3). Results are presented as Mean ± SD. * *p* < 0.05, ** *p* < 0.01, NS: Not Statistically, for the statistical analysis of Lac vs. Con.

**Figure 4 ijms-23-13996-f004:**
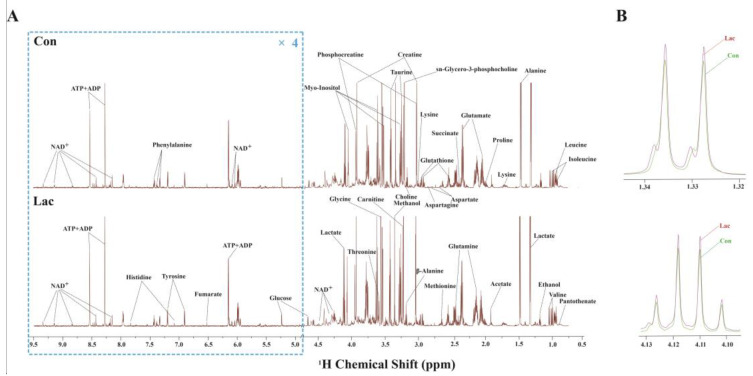
Typical 1D ^1^H spectra of aqueous extracts derived from the two groups of myoblasts. (**A**) 1D ^1^H spectra of controls (Con) and lactate-treated cells (Lac) were recorded on an 850 MHz NMR spectrometer at 25 °C. Vertical scales are kept constant in all the spectra, and the resonance region of water (4.75–4.85 ppm) was removed. For the purpose of clarity, the resonance region of 4.85–9.5 ppm has been magnified four times compared with the other region of 0.75–4.75 ppm. Abbreviations: NAD^+^, nicotinamide adenine dinucleotide; AXP, adenine mono/di/triphosphate; and (**B**) amplified resonance regions of lactate in Con cells (green) and Lac cells (red).

**Figure 5 ijms-23-13996-f005:**
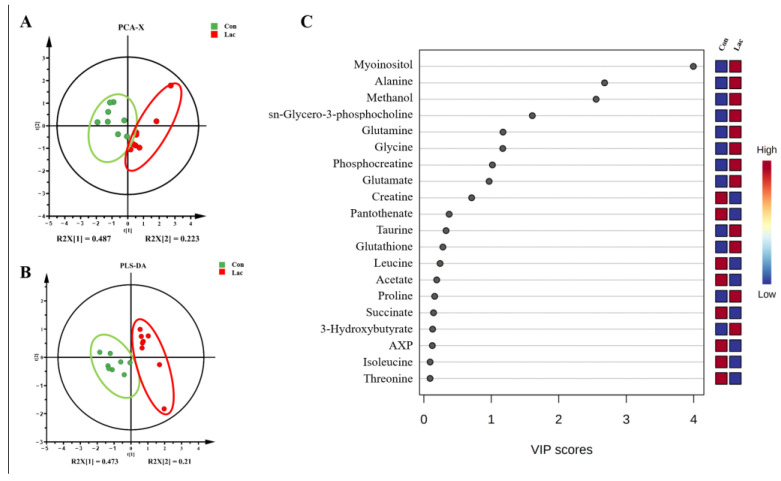
Multivariate statistical analysis for the NMR data of the two groups of myoblasts. (**A**) PCA scores plot for the non-supervised multivariate statistical analysis of Lac and Con; (**B**) PLS-DA scores plot for the supervised multivariate statistical analysis of Lac vs. Con; and (**C**) VIP scores-ranking plot of significant metabolites screened from the PLS-DA model with the criterion of VIP > 1.

**Figure 6 ijms-23-13996-f006:**
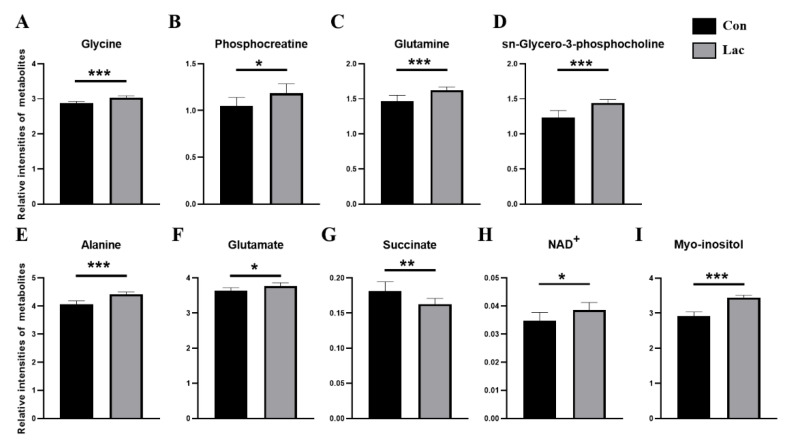
Quantitative comparison of relative levels of differential metabolites between the two groups of myoblasts. (**A**) Glycine (**B**) Phosphocreatine (**C**) Glutamine (**D**) sn-Glycero-3-phosphocholine (**E**) Alanine (**F**) Glutamate (**G**) Succinate (**H**) NAD+ (**I**) Myo-inositol. * *p* < 0.05, ** *p* < 0.01, *** *p* < 0.001. *n* = 8 for each group.

**Figure 7 ijms-23-13996-f007:**
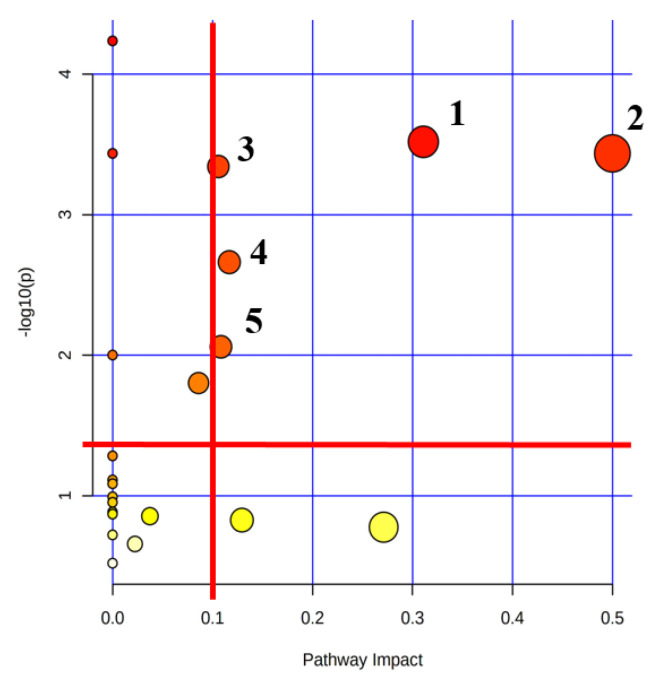
The metabolic pathway analysis of lactate-treated myoblasts relative to controls. Five significantly altered metabolic pathways were identified with two criteria of pathway impact values > 0.1 and *p* values < 0.05 (i.e., −log10(*p*) > 1.3), including (1) alanine, aspartate and glutamate metabolism; (2) D-Glutamine and D-Glutamate metabolism; (3) glyoxylate and dicarboxylate metabolism; (4) arginine biosynthesis; and (5) glutathione metabolism.

**Figure 8 ijms-23-13996-f008:**
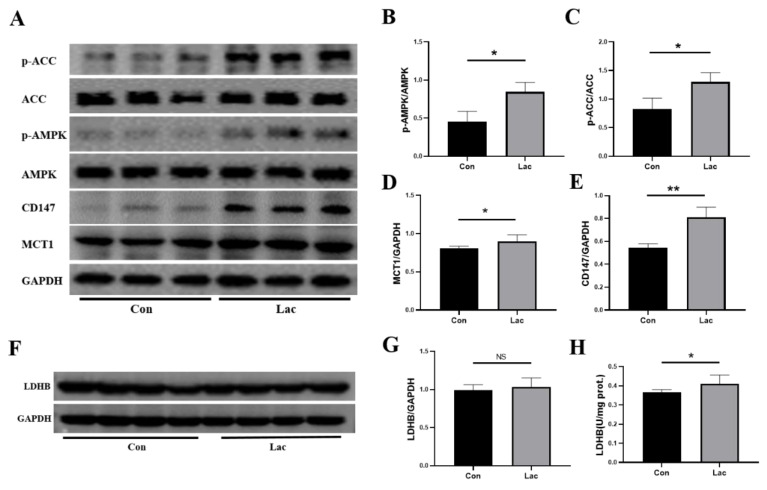
Lactate upregulated energy metabolism in myoblasts. Cells were treated for 2 h in either DMEM medium (Con) or DMEM medium supplemented with 5 mM lactate (Lac). (**A**) Expressions of p-AMPK (Thr172), p-ACC (Ser79), CD147 and MCT1 LDHB analyzed by Western blot (*n* = 3); (**B**–**E**) Gray value analyses corresponding to Panel (**A**); (**F**) LDHB analyzed by Western blot (*n* = 4); (**G**) Gray value analyses corresponding to Panel (**F**); and (**H**) Enzymatic activity of LDHB (*n* = 5). Results are presented as Mean ± SD. * *p* < 0.05, ** *p* < 0.01, NS: Not Statistically, for the statistical analysis of Lac vs. Con.

**Figure 9 ijms-23-13996-f009:**
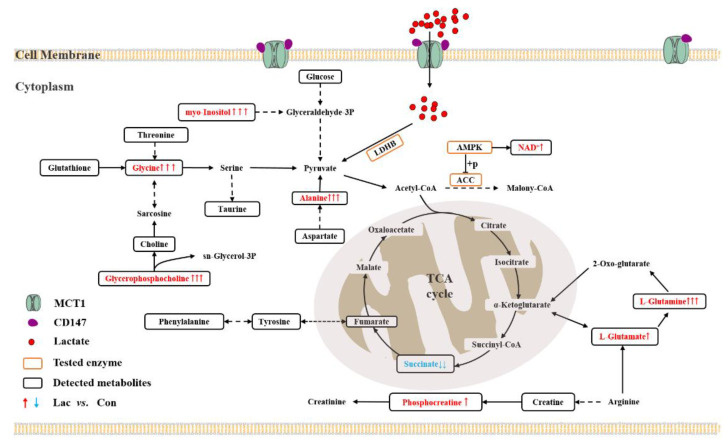
Overview of the metabolic mechanisms accounting for the effects of lactate treatment on the proliferation and differentiation of myoblasts based on KEGG metabolic pathways. The dotted arrow indicates multiple biochemical reactions; the solid arrow denotes a single biochemical reaction. The metabolites without frame refer to undetected metabolites.

## Data Availability

Not applicable.

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
