# Peer review of "Lactate Activates AMPK Remodeling of the Cellular Metabolic Profile and Promotes the Proliferation and Differentiation of C2C12 Myoblasts"

_ijms, 2022, doi:10.3390/ijms232213996_

Round 1
Reviewer 1 Report
In this study, Zhou et al show that Lactate is able to promote proliferation and differentiation of C2C12 myoblasts by activates AMPK signaling pathway. Particularly, by using NMR-based metabonomic profiling, they authors demonstrated the enhanced expression levels of proteins related to cellular proliferation and differentiation induced by lactate. In general, I find this work to be well designed and results clearly presented, and have important significance in understanding the role of lactate as a metabolic regulator. Some minor revisions are recommended. Please see the detailed comments below.
1. Some experimental details need to be provided.
a. In session 4.1, provide more information about the inverted microscope used and the imaging parameters.
b. Session 4.4, a brief description of the protocol and control should be included.
c. Session 4.5, how much total protein was loaded onto SDS-PAGE.
d. Session 4.5, provide more information about the secondary antibody as well as the incubation time of antibody.
e. Session 4.6, provide more details about the freeze dry procedure.
f. Provide more information about the water suppression pulse.
g. provide more information how water peak was removed from the spectrum
2. The concentration of lactate used in the study is 5 mM. How was that dose determined? Have the authors tried the dose rep
3. There are other factors/pathways that could have contributed to the proliferation and viability of the cells, how did the authors rule out other possible factors?
4. Fig. 2, it would be helpful to add a cell counts vs. date plot.
5. Scale bar missing in B.
6. Fig. 4, x-axis labels are needed.
Author Response
Comments and Suggestions for Authors
In this study, Zhou et al show that Lactate is able to promote proliferation and differentiation of C2C12 myoblasts by activates AMPK signaling pathway. Particularly, by using NMR-based metabonomic profiling, they authors demonstrated the enhanced expression levels of proteins related to cellular proliferation and differentiation induced by lactate. In general, I find this work to be well designed and results clearly presented, and have important significance in understanding the role of lactate as a metabolic regulator. Some minor revisions are recommended. Please see the detailed comments below.
Q1. Some experimental details need to be provided.
- In session 4.1, provide more information about the inverted microscope used and the imaging parameters.
- Session 4.4, a brief description of the protocol and control should be included.
- Session 4.5, how much total protein was loaded onto SDS-PAGE.
- Session 4.5, provide more information about the secondary antibody as well as the incubation time of antibody.
- Session 4.6, provide more details about the freeze dry procedure.
- Provide more information about the water suppression pulse.
- provide more information how water peak was removed from the spectrum
A1. Thanks for your constructive comments and suggestions. We have made modifications following these comments.
- In session 4.1, we have provided more information about the inverted microscope used and the imaging parameters (line 358).
- In session 4.4, we have added a brief description of the protocol and control (line379-381).
- In session 4.5, we have stated the amount of total protein loaded onto SDS-PAGE (line 389).
- In session 4.5, we have provided more information about the secondary antibody as well as the incubation time of antibody (line 397-398).
- In session 4.6, we have provided more details about the freeze dry procedure (line 405-408).
- We have provided more information about the water suppression pulse (line 415-417).
- We have provided more information how water peak was removed from the spectrum (425-428).
Q2. The concentration of lactate used in the study is 5 mM. How was that dose determined? Have the authors tried the dose rep
A2. Thanks for your comment. Considering the physiological range of lactate concentration in blood is around 5 mM after a certain time of mid-intensity aerobic exercise in human, we chose 5 mM as the concentration of lactate for treating C2C12 cells1,2. We have added one sentence in the revised manuscript (line 90-93).
Q3. There are other factors/pathways that could have contributed to the proliferation and viability of the cells, how did the authors rule out other possible factors?
A3. It is true that there are other factors/pathways potentially contributing to the proliferation and viability of the cells. Our results indicated that lactate supplement could activate AMPK, and inhibition of AMPK signaling using dorsomorphin (the inhibitor of AMPK) significantly blocked the promotive effect of lactate supplementation on cellular proliferation and viability of the cells (the newly added Figure S3 in Supplementary Materials). Thus, the present study primarily focused on the activation of AMPK signaling induced by lactate supplement, which served as a key pathway.
Q4. Fig. 2, it would be helpful to add a cell counts vs. date plot.
A4. Thanks for your constructive suggestion. The pictures shown in Fig. 2 were taken in the process of differentiation where cell fusion had started. It is more difficult to count cell numbers in the process of differentiation.
Q5. Scale bar missing in B.
A5. Thanks for your comment. We have added the scale bar to Fig. 2B in the revised manuscript (Line 120).
Q6. Fig. 4, x-axis labels are needed.
A6. Thanks for your suggestion. We have supplemented the X-axis to Fig 4 in the revised manuscript (Line 147).
References
1 Philp, A., Macdonald, A. L. & Watt, P. W. Lactate--a signal coordinating cell and systemic function. J Exp Biol 208, 4561-4575, doi:10.1242/jeb.01961 (2005).
2 Fattor, J. A., Miller, B. F., Jacobs, K. A. & Brooks, G. A. Catecholamine response is attenuated during moderate-intensity exercise in response to the "lactate clamp". Am J Physiol-Endoc M 288, E143-E147, doi:10.1152/ajpendo.00117.2004 (2005).

Reviewer 2 Report
The manuscript indicated that lactate activated AMPK signaling pathway, cell proliferation, and cell differentiation, and changes intracellular metabolism in C2C12 cells. Recently, it is known that lactate acts as a signaling molecule in several organ and cells, while the authors focused on the effects of lactate on cell morphology, metabolism, and signaling pathway in muscle cells. Their obtained data seems to be novel finding in myoblast, however, there is still unclear the relationship among AMPK signaling, ERK signaling, cell morphology, and intracellular metabolism in the author’s experiment. The reviewer consider they should perform additional experiments especially in AMPK signaling pathway to demonstrate their proposed possibilities.
1. The authors indicated AMPK signaling is affected by lactate supplementation in C2C12. They state some possibilities that the signaling pathway relates to lactate-induced alteration in intracellular metabolism and cell morphology. Since AMPK is linked to several metabolic pathways, the possibility is reliable. However, they should confirm and demonstrate the possibility using the inhibitor like Compound C.
2. In cell differentiation experiment, the authors investigated p38, and myogenic regulatory factors (MRFs). Their results indicated that the lactate enhanced the expression of MRFs, but not p38 phosphorylation. From these results, p38-MAPK signaling pathway does not relate to cell differentiation?? Please discuss about this point.
3. In cell proliferation experiment, they indicated lactate activated ERK and Akt. The activation is regulated by AMPK or not??
4. In NMR-based metabolome analysis, the authors indicated the metabolites of TCA cycle is decreased by lactate supplementation, based on succinate result. However, as shown in supplemental data, fumarate content is not changed by the supplementation. The reviewer consider other additional data is required to indicate the effect on TCA cycle.
5. They indicated that glycine, alanine, glutamine, and glutamate concentrations were increased by lactate supplementation in C2C12 cells. These amino acids were incorporated from culture medium or synthesized by intracellular metabolism??
6. Regarding to manuscript title. AMPK is highlighted in the present title, however, they should re-consider about title because of lack of evidences that AMPK related to cell differentiation and intracellular metabolism.
7. In Line 154, they stated that lactate was removed from data analysis, however, in Fig5C, there is lactate data. I was confused this point.
Author Response
Comments and Suggestions for Authors
The manuscript indicated that lactate activated AMPK signaling pathway, cell proliferation, and cell differentiation, and changes intracellular metabolism in C2C12 cells. Recently, it is known that lactate acts as a signaling molecule in several organ and cells, while the authors focused on the effects of lactate on cell morphology, metabolism, and signaling pathway in muscle cells. Their obtained data seems to be novel finding in myoblast, however, there is still unclear the relationship among AMPK signaling, ERK signaling, cell morphology, and intracellular metabolism in the author’s experiment. The reviewer consider they should perform additional experiments especially in AMPK signaling pathway to demonstrate their proposed possibilities.
Q1. The authors indicated AMPK signaling is affected by lactate supplementation in C2C12. They state some possibilities that the signaling pathway relates to lactate-induced alteration in intracellular metabolism and cell morphology. Since AMPK is linked to several metabolic pathways, the possibility is reliable. However, they should confirm and demonstrate the possibility using the inhibitor like Compound C.
A1. Thank you for your constructive comment and suggestion. We have confirmed the possibility using dorsomorphin (Compound C, the AMPK inhibitor). We observed that the inhibition of AMPK signaling significantly blocked the promotive effect of lactate supplementation on cell proliferation (the newly added Figure S3). We have added several sentences in the revised manuscript (line 216-218).
Q2. In cell differentiation experiment, the authors investigated p38, and myogenic regulatory factors (MRFs). Their results indicated that the lactate enhanced the expression of MRFs, but not p38 phosphorylation. From these results, p38-MAPK signaling pathway does not relate to cell differentiation?? Please discuss about this point.
A2. Thanks for your comment. It is known that p38-MAPK signaling relates to cell differentiation1-3,In the early state of differentiation, activation of p38-MAPK signaling can upregulate p21, which will lead to limitation of cell cycle, make myogenic cells exit cell cycle and enter terminal differentiation1-3. We did not observe a significantly promotive effect of lactate supplementation on the expression of phosphorylated p38, probably as we collected cell samples in the late state of differentiation.
Q3. In cell proliferation experiment, they indicated lactate activated ERK and Akt. The activation is regulated by AMPK or not??
A3. Thanks for your comment. Previous studies have demonstrated that the activation of ERK4 and akt 5,6 can promote the cellular proliferation. In the present study, we measured the expressions of p-ERK and p-akt to reflect the promoted cell proliferation. Actually, it is unclear that whether the activation of ERK and Akt is regulated by AMPK. This issue is worthy to be further addressed in future. We have modified several sentences relevant to the activation of ERK and Akt in the revised manuscript (line 98-101).
Q4. In NMR-based metabolome analysis, the authors indicated the metabolites of TCA cycle is decreased by lactate supplementation, based on succinate result. However, as shown in supplemental data, fumarate content is not changed by the supplementation. The reviewer consider other additional data is required to indicate the effect on TCA cycle.
A4. Thanks for your constructive comment and suggestion. Besides detecting concentrations of metabolites in TCA cycle, we have also detected activities of several metabolic enzymes in TCA cycle, including Isocitrate dehydrogenase (IDH), malate dehydrogenase (MDH) and succinate dehydrogenase (SDH). The results showed that lactate supplement significantly enhanced the enzymatic activities of IDH and MDH (the newly added Figure S4), indicating the effect of lactate supplement on TCA cycle. We have added several sentences in the revised manuscript (line 314-317).
Q5. They indicated that glycine, alanine, glutamine, and glutamate concentrations were increased by lactate supplementation in C2C12 cells. These amino acids were incorporated from culture medium or synthesized by intracellular metabolism??
A5. Thank you for your comment. Our study showed the increased concentrations of glycine, alanine, glutamine, and glutamate in C2C12 cells. Considering that the culture medium did not contain alanine and glutamate, the increased concentrations of these two amino acids might be synthesized by intracellular metabolism. Additionally, glycine and glutamine act as important substrates for the synthesis of other amino acids, their increased concentrations in cells were potentially contributed by culture medium.
Q6. Regarding to manuscript title. AMPK is highlighted in the present title, however, they should re-consider about title because of lack of evidences that AMPK related to cell differentiation and intracellular metabolism.
A6. Thanks for your comment. It has been reported that the activation of AMPK signaling can influence cellular metabolism, further activating muscle satellite cell7-11. Our results indicated that lactate supplementation can activate AMPK, significantly remodeling metabolism of myoblasts. Furthermore, we observed that the inhibition of AMPK signaling using the inhibitor dorsomorphin distinctly blocked the promotive effect of lactate supplementation on cell proliferation (Figure S3), which provides an evidence that AMPK related to cell proliferation and intracellular metabolism.
Q7. In Line 154, they stated that lactate was removed from data analysis, however, in Fig5C, there is lactate data. I was confused this point.
A7. Thanks for your comment. We have fixed this mistake by excluding the lactate data from Fig. 5C in the revised manuscript (line 168).
References
1 Keren, A., Tamir, Y. & Bengal, E. The p38 MAPK signaling pathway: a major regulator of skeletal muscle development. Mol Cell Endocrinol 252, 224-230, doi:10.1016/j.mce.2006.03.017 (2006).
2 Lluis, F., Perdiguero, E., Nebreda, A. R. & Munoz-Canoves, P. Regulation of skeletal muscle gene expression by p38 MAP kinases. Trends in Cell Biology 16, 36-44, doi:10.1016/j.tcb.2005.11.002 (2006).
3 Wu, Z. G. et al. p38 and extracellular signal-regulated kinases regulate the myogenic program at multiple steps. Molecular and Cellular Biology 20, 3951-3964, doi:Doi 10.1128/Mcb.20.11.3951-3964.2000 (2000).
4 Ohno, Y. et al. Lactate increases myotube diameter via activation of MEK/ERK pathway in C2C12 cells. Acta Physiol (Oxf) 223, e13042, doi:10.1111/apha.13042 (2018).
5 Bodine, S. C. et al. Akt/mTOR pathway is a crucial regulator of skeletal muscle hypertrophy and can prevent muscle atrophy in vivo. Nat Cell Biol 3, 1014-1019, doi:DOI 10.1038/ncb1101-1014 (2001).
6 Elia, D., Madhala, D., Ardon, E., Reshef, R. & Halevy, O. Sonic hedgehog promotes proliferation and differentiation of adult muscle cells: Involvement of MAPK/ERK and PI3K/Akt pathways. Biochim Biophys Acta 1773, 1438-1446, doi:10.1016/j.bbamcr.2007.06.006 (2007).
7 Theret, M. et al. AMPK alpha 1-LDH pathway regulates muscle stem cell self-renewal by controlling metabolic homeostasis. Embo J 36, 1946-1962, doi:10.15252/embj.201695273 (2017).
8 Hardie, D. G., Ross, F. A. & Hawley, S. A. AMPK: a nutrient and energy sensor that maintains energy homeostasis. Nat Rev Mol Cell Bio 13, 251-262, doi:10.1038/nrm3311 (2012).
9 Feldman, J. L. & Stockdale, F. E. Skeletal muscle satellite cell diversity: satellite cells form fibers of different types in cell culture. Dev Biol 143, 320-334, doi:10.1016/0012-1606(91)90083-f (1991).
10 Le Grand, F. & Rudnicki, M. A. Skeletal muscle satellite cells and adult myogenesis. Curr Opin Cell Biol 19, 628-633, doi:10.1016/j.ceb.2007.09.012 (2007).
11 Mauro, A. Satellite cell of skeletal muscle fibers. J Biophys Biochem Cytol 9, 493-495, doi:10.1083/jcb.9.2.493 (1961).

Round 2
Reviewer 2 Report
The authors added the additional data such as AMPK inhibitor work and enzyme assay of TCA cycle, which were requested in the previous review process.
They also modified the sentences especially in discussion section.
I consider this manuscript is acceptable for the publication.